# Deforestation and human proximity influence *Trypanosoma cruzi* infection in palm-dwelling triatomines

Gabriel Z. Laporta[1]*, Leandro J. Ramos[2], Carla M. Santana[3], Wandercleyson U. Abreu[3], Roberto C. Ilacqua[1], Melissa S. Nolan[4,5], Andreia F. Brilhante[2], Fernanda P. Madeira[6], Dionatas U. O. Meneguetti[2], Fredy Galvis-Ovallos[7], Paula R. Prist[8], Denis Valle[9], Marcia A. Sperança[3]

**1** Graduate Program in Health Sciences, FMABC University Center, Santo André, São Paulo, Brazil, **2** Center for Health Sciences and Sports, Federal University of Acre, Rio Branco, Acre, Brazil, **3** Center for Natural and Human Sciences, Federal University of ABC, São Bernardo do Campo, São Paulo, Brazil, **4** Department of Epidemiology and Biostatistics, Arnold School of Public Health, University of South Carolina, Columbia, South Carolina, United States of America, **5** Institute for Infectious Disease Translational Research, University of South Carolina, Columbia, South Carolina, United States of America, **6** Multidisciplinary Center, Forest Campus, Federal University of Acre, Cruzeiro do Sul, Acre, Brazil, **7** Department of Epidemiology, School of Public Health, University of São Paulo, São Paulo, São Paulo, Brazil, **8** Forests and Grasslands Programme, IUCN, Washington, District of Columbia, United States of America, **9** School of Forest, Fisheries, and Geomatics Sciences, University of Florida, Gainesville, Florida, United States of America

* gabriel.laporta@fmabc.br

## Abstract

In the southwestern Brazilian Amazon, palm-dwelling triatomines maintain sylvatic transmission cycles of *Trypanosoma cruzi*, the causative agent of Chagas disease, and *Trypanosoma rangeli*, a related non-pathogenic parasite. Deforestation can reduce biodiversity and increase pathogen prevalence in triatomine populations; however, the effects of landscape structure on triatomine infection patterns remain poorly understood. Here, we address this knowledge gap by examining how forest cover and proximity to human dwellings influence triatomine infection patterns across gradients of deforestation. Field surveys were conducted in 2022 and 2024 in 20 landscape units in Cruzeiro do Sul, Acre state, Brazil, where triatomines were collected from palm trees located at varying distances from inhabited households. Distances and land-use composition were quantified from high-resolution drone and satellite imagery, while parasite infections were identified using molecular assays. Bayesian binomial mixed-effects models revealed contrasting responses between parasites. *T. cruzi* infection probability was higher in more deforested landscapes and was further modulated by palm-household distance, with the strongest effects observed for palms closer to dwellings. In contrast, *T. rangeli* infection showed no supported association with forest cover or distance to households. Blood meal analysis revealed frequent feeding on sylvatic hosts, particularly marsupials, and detected human blood in a nymph collected only 33 m from a household; *T. cruzi* infections detected in the study

**Data availability statement:** All relevant data are within the manuscript and its Supporting Information files.

**Funding:** This study was funded by the São Paulo Research Foundation (FAPESP) through grants 2021/06669-6 (awarded to GZL) and 2022/10392-2 (awarded to MAS) (https://fapesp.br); by the National Council for Scientific and Technological Development (CNPq) through grant 307293/2023-8 (awarded to GZL), grant 401549/2023-2 under the Amazônia Initiative Program +10 (awarded to AFB), and FAPAC grant 0043014464/00032/2022-34, also associated with the Amazônia Initiative Program +10 (awarded to AFB) (https://www.gov.br/cnpq); and by the USC Infectious Disease Translational Research Center (awarded to MSN) (https://sc.edu/about/centers_institutes/infectious-disease-translational-research/). RCI received a doctoral scholarship from the São Paulo Research Foundation (FAPESP) (grant 2023/08053-8). CMS received a graduate scholarship from the Coordenação de Aperfeiçoamento de Pessoal de Nível Superior (CAPES) (https://www.gov.br/capes). The funders had no role in the study design, data collection and analysis, decision to publish, or preparation of the manuscript.

**Competing interests:** The authors have declared that no competing interests exist.

were exclusively assigned to TcI discrete typing unit, a lineage commonly associated with sylvatic transmission. These findings demonstrate that deforestation reshapes host-vector-parasite interactions in palm-based systems, increasing spillover risk at the sylvatic-human interface without requiring domiciliated triatomines.

## Introduction

Triatomines are hematophagous insects that act as vectors of *Trypanosoma cruzi*, the etiological agent of Chagas disease, and *Trypanosoma rangeli*, a non-pathogenic parasite frequently found in the same vectors [1–3]. *Trypanosoma rangeli* is primarily associated with triatomines of the genus *Rhodnius*, in which it can complete its developmental cycle without causing disease in humans [4]. Despite its non-pathogenic nature, *T. rangeli* is epidemiologically relevant because it can co-occur with *T. cruzi* in vectors and biological samples, potentially interfering with diagnostic assays and leading to cross-reactivity or false-positive results depending on the detection method used [5,6]. In the Brazilian Amazon, Chagas disease remains a neglected zoonosis that predominantly manifests through oral transmission, following the ingestion of food contaminated with *T. cruzi*-infected triatomine feces or secretions [7–9]. Although acute Chagas disease (ACD) can be effectively treated with nifurtimox or benznidazole when diagnosed early, most infections occur in remote regions with limited access to early diagnosis and treatment efficacy [10,11].

Deforestation is a major ecological driver of vector-borne and zoonotic diseases in the Amazon [12–14]. Landscape alteration modifies vector habitats, reshapes host communities, and alters the interfaces between sylvatic and human environments [15]. In the context of Chagas disease, forest loss has been associated with increased invasion of households by sylvatic triatomines seeking alternative blood sources following habitat degradation [1,14]. In Acre, Rondônia, and Pará states, species such as *Rhodnius montenegrensis*, *Rhodnius robustus*, and *Panstrongylus geniculatus* have been reported in peridomestic settings, frequently infected with *T. cruzi* or *T. rangeli* [7,16,17]. These observations suggest that deforestation gradients and landscape fragmentation influence both the spatial distribution of triatomines and their infection profiles.

Palm trees play a crucial ecological role as nesting habitats for several triatomine species in the Amazon [18–20]. Palm crowns constitute stable microhabitats that support diverse communities of vertebrate reservoirs, including marsupials and rodents, which sustain transmission cycles of both *T. cruzi* and *T. rangeli* [21,22]. We therefore hypothesize that variation in forest cover and proximity to human dwellings modulates triatomine infection risk. Specifically, *T. cruzi* infections are expected to be more frequent in highly anthropized landscapes, whereas *T. rangeli* is anticipated to be more common in preserved forest habitats (Fig 1).

Identifying how landscape structure shapes parasite-vector associations is critical for assessing Chagas disease risk and informing prevention strategies [23–25]. In this study, we evaluate the relationship between forest cover, distance to households,

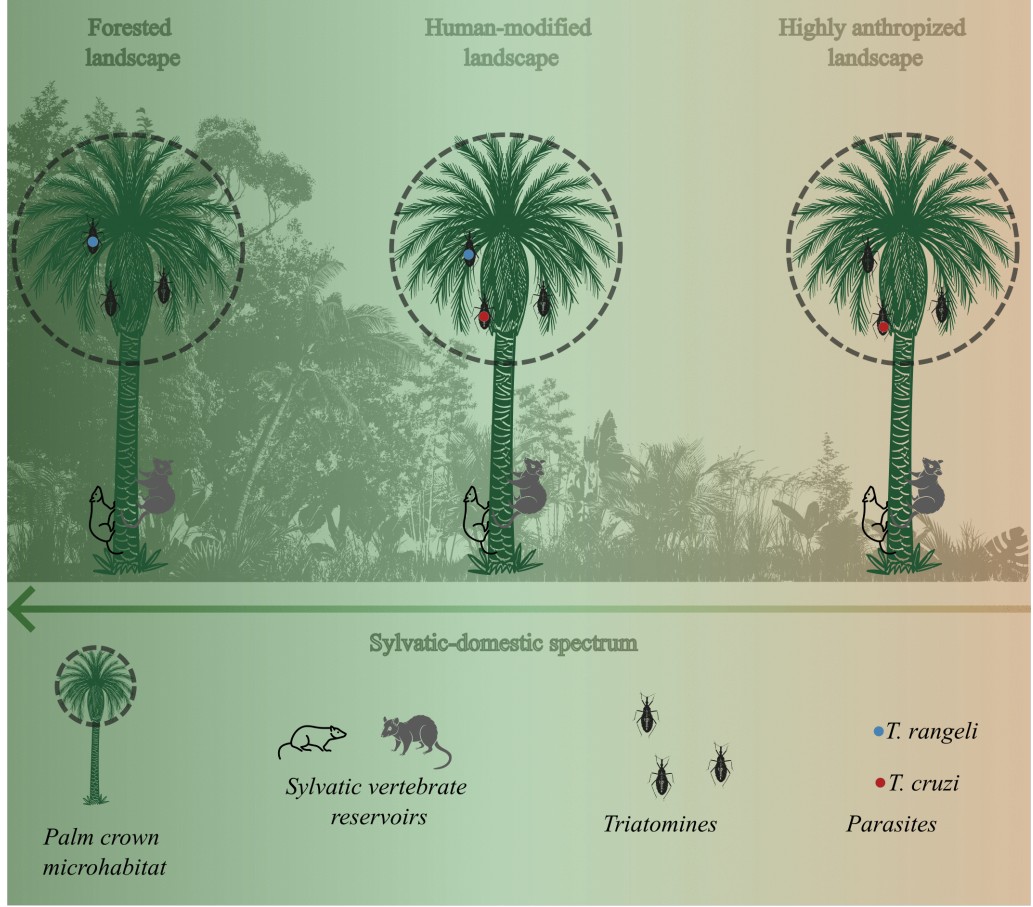

**Fig 1. Conceptual diagram of the triatomine microcosm in Amazonian palm trees across a sylvatic-domestic gradient.** Palm crowns provide microhabitats that support sylvatic vertebrate hosts and palm-dwelling triatomine colonies. The diagram depicts three palm-based landscapes along a deforestation and human-use gradient: forested landscapes, where triatomines are primarily associated with *T. rangeli*; human-modified landscapes, where both *T. rangeli* and *T. cruzi* may occur; and highly anthropized landscapes near human dwellings, where *T. cruzi* can predominate. The figure was created by the authors using Inkscape v1.4, with icons adapted from the Noun Project.

and infection by *T. cruzi* and *T. rangeli* in palm-dwelling triatomines across landscapes of Cruzeiro do Sul, Acre state. Using ecological field data collected in 2022 and 2024 and Bayesian statistical models, we test whether deforestation and proximity to human dwellings increase the probability of *T. cruzi* infection, while more preserved forest environments favor *T. rangeli*.

## Materials and methods

### Study area

The study was conducted in the municipality of Cruzeiro do Sul, located in the westernmost region of Acre State, Brazil (Fig 2). The municipality lies within the southwestern Amazon and is currently experiencing rapid forest loss and expansion of rural settlements. Deforestation in Cruzeiro do Sul is a relatively recent process, driven by political and socio-economic transformations, making the area suitable for investigating how landscape change influences sylvatic disease transmission [26–28].

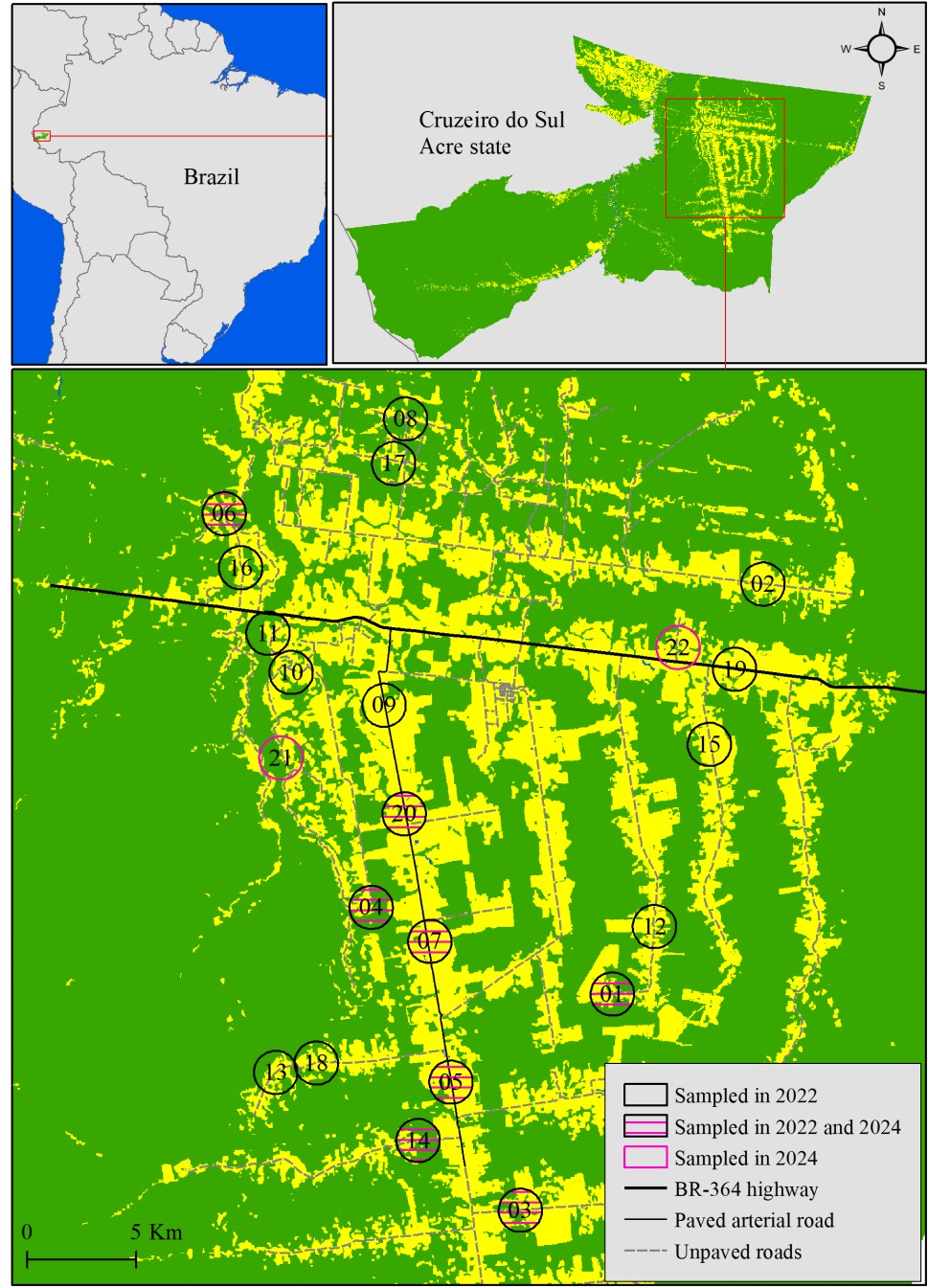

**Fig 2. Study area and landscape context of triatomine sampling in Cruzeiro do Sul, Acre, Brazil.** Land-use and land-cover map showing forested (green) and anthropized areas (yellow). Circles represent the 22 landscape units surveyed for triatomines. Solid black circles indicate landscapes sampled in 2022; solid pink circles indicate landscapes sampled in 2024, and combined black-and-pink circles indicate landscapes sampled in both 2022 and 2024. The map was created by the authors using ArcMap 10.8.2 based on open-access datasets from IBGE (administrative boundaries) and MapBiomas (land-use and land-cover), along with field-collected GPS data.

The region is a known focus of ACD transmitted orally, with approximately one to five outbreaks reported annually, affecting 5–20 individuals [3]. Diagnostic and treatment protocols for Chagas disease are available in local health services, and the prevalence of untreated asymptomatic or chronic infection is estimated at around 1% [8,29].

**Landscape site selection and geospatial metrics**

Twenty landscape sites (landscapes 1–20 in Fig 2) were selected to represent a gradient of forest cover and anthropogenic disturbance [30]. Site selection was guided by spatial modeling integrating forest cover, deforestation timelines, proximity to human settlements, and projected deforestation probabilities for 2025 [31,32]. A random forest classifier combined with D-optimality-based random selection ensured a representative and spatially balanced sampling design [32,33]. Each site corresponded to a rural settlement lot typically used for cassava cultivation and small-scale cattle grazing. The regional environment consists of tropical moist forests, gentle slopes (200–300 m elevation), dystrophic yellow argisol soils, and a high water table that ensures perennial water availability [34,35].

At each landscape site, one dominant palm tree (*Attalea* sp.) located in the vicinity of a household was selected for triatomine sampling. Palms were chosen based on a combination of proximity to human dwellings and structural suitability for triatomines, prioritizing well-developed crowns with accumulated organic material known to support triatomine colonies. Because landscape sites were selected to represent a gradient of forest cover and anthropogenic disturbance, some sites contained no suitable palms in 2022 (landscapes 02, 08, 09, 12, and 13 in Fig 2). Where palms were present, trees at varying distances from households were sampled. In 2024, a subset of sites was revisited to intentionally sample palms located farther from human residences, including palms near forest edges or within interior forest fragments, in order to better capture distance-related variation. Two additional landscapes containing suitable palms were included in 2024 to increase sampling coverage.

All landscape sites and sampled palms were georeferenced using GPS+GLONASS (Garmin International, Olathe, Kansas, USA) under UTM WGS 1984 Zone 19S projection. High-resolution Sentinel-2 images (10 m) from August 2022 and July 2024 were obtained from the Copernicus Open Access Hub (European Space Agency, Paris, France). Images were preprocessed and classified in QGIS 3.3 using the Semi-Automatic Classification Plugin [31,36,37], mapping four land-cover classes: native forest, secondary forest, exposed soil, and surface water [23]. Forest cover was quantified as the proportion of native forest within a 3-km$^2$ circular buffer surrounding each household. This buffer size was selected to reflect the local spatial scale relevant to palm-dwelling *Rhodnius* spp., whose dispersal is typically limited to short-range flights among nearby palms and host habitats [38–40].

High-resolution RGB orthomosaics of each landscape site were generated from drone imagery acquired in July 2024 using a Mavic 3 Multispectral (DJI, China) and processed in DroneDeploy. Flights were planned to cover each landscape site with sufficient overlap for accurate orthorectification. The resulting orthomosaics (few centimeters per pixel) enabled precise georeferencing of all sampled palms, calculation of Euclidean distances to the human domicile, and verification of palm locations sampled in 2022.

**Triatomine collection and processing**

Within each sampled palm tree, triatomines were collected following established entomological protocols for sylvatic capture [40,41]. Trees were felled using a chainsaw (MS 382; Stihl, Waiblingen, Germany) under environmental authorization from the Brazilian Ministry of the Environment (license n. 52260−1). Bracts and palm fibers were manually inspected for triatomines, which were collected alive, placed in 50 mL Falcon tubes, and kept in thermal boxes for transport to the field laboratory. A supplementary video illustrates triatomine collection from palm bracts (S1 Video).

Collected specimens were morphologically identified to genus level (*Rhodnius*) using taxonomic keys [40,41]. The intestinal contents of adult triatomines and selected nymphs were diluted in saline and examined microscopically for flagellate protozoa. All specimens, including their digestive contents, were individually preserved in microtubes with 80%

ethanol at −20 °C for subsequent molecular detection of *T. cruzi* and *T. rangeli* and for molecular identification of the ingested blood.

## Molecular diagnosis

Triatomines preserved in 80% ethanol were washed twice with 1x PBS (137 mM NaCl, 2.7 mM KCl, 10 mM $Na_2HPO_4$, and 1.8 mM $KH_2PO_4$) to remove residual ethanol. Each specimen was incubated in 200–500 µL of lysis buffer containing proteinase K at 56 °C for 16 h [42]. After digestion, 200 µL of lysate was extracted using Trizol reagent followed by chloroform and ethanol precipitation steps (centrifugations at 14,000xg, 4 °C). Pellets were washed with 70% ethanol, dried, and resuspended in 40 µL of 10 mM Tris-HCl (pH 7.5); nucleic acids were stored at −20 °C.

Detection of *T. cruzi* was performed by qPCR targeting the satellite DNA region [5,6] using primers T.cruzi1 (5'-ASTCGGCTGATCGTTTTCGA-3'), T.cruzi2 (5'-AATTCCTCCAAGCAGCGGATA-3'), and probe T.cruzi3 (FAM-5'-CACACACTGGACACCAA-3'-BHQ1). Trypanosome species were identified by PCR amplification of a 661-bp fragment of the 18S rRNA gene using oligonucleotides designed with available bioinformatics software [43,44]. *T. cruzi*-positive samples were genotyped by multiplex qPCR to determine discrete typing unit (DTU) [45].

The origin of ingested blood was determined by PCR. Human blood was identified via the mitochondrial hypervariable region 1 (HV1; 440 bp) [46], and other vertebrate hosts were identified using a 307 bp fragment of the cytochrome b (cytb) gene [47]. PCR products were visualized on 1.5% agarose gels, purified with ExoSAP, and sequenced using BigDye Terminator 3.1 chemistry on an ABI 3500 XL Genetic Analyzer. Sequences were identified via BLAST searches.

## Statistical analysis and modeling

To test hypotheses about landscape influences on triatomine infection, we used a Bayesian generalized linear mixed model implemented in JAGS via the jagsUI package in R [48]. The response variable was the number of infected triatomines per palm ($y_i$), modeled as a binomial outcome with the total number of triatomines collected ($N_i$) as the number of trials:

$$y_i \sim Binomial(p_i, N_i)$$

The probability of infection $p_i$ was modeled on the logit scale as a linear combination of standardized fixed effects (distance from household, proportion of forest within a 3 $km^2$ buffer, and their interaction when tested) and a random intercept for each landscape site to account for repeated measures:

$$logit(p_i) = b_{0,\ site[i]} + \sum_j \beta_j x_{ij}$$

Weakly informative priors were assigned to the fixed-effect coefficients $\beta_j \sim$ Normal (0,1), while random intercepts were drawn from a normal distribution with hyperparameters $b_{0,site[i]} \sim$ Normal ($\mu$, $\sigma^2$). Hyperpriors were specified as $\mu_0 \sim$ Normal (0,10) and $\sigma \sim$ Uniform (0,10).

Posterior distributions were obtained using three MCMC chains, with 10,000 iterations, a burn-in of 1,000, and thinning only kept samples every five iterations. Model convergence was assessed via the $R$ statistic. Probabilities of infection along gradients of distance and forest cover were derived by back transforming from the logit scale for visualization.

## Ethics statement

This study did not involve human subjects. Access to triatomine genetic material and *Trypanosoma* DNA complied with regulations of the Brazilian Ministry of the Environment (SISGEN registration n. AFF4128, A910886). Triatomine

collections were authorized by the Brazilian Ministry of the Environment (SISBIO n. 95072−1), including specific permission for palm tree cutting during sampling (authorization n. 52260−1).

## Results

Across the 15 landscape sites with palms in 2022, a total of 23 triatomines were collected (Table 1). In 2024, a subset of previously sampled landscapes was revisited, and two additional landscapes with suitable palms were included, resulting in 32 triatomines collected. All specimens were morphologically identified to genus *Rhodnius* (S1 Fig).

Molecular screening confirmed the presence of *T. cruzi* and *T. rangeli* in both sampling years (S1 Table). All *T. cruzi*-positive samples were assigned to DTU TcI. Infections occurred in adults and nymphs, reinforcing evidence of active transmission cycles associated with palms.

Blood meal analysis showed a predominance of sylvatic hosts, with marsupials (*Philander opossum* and *Didelphis marsupialis*) frequently detected at sites where trypanosome infections occurred (S1 Table). Blood from lizards and domestic or peri-domestic animals was also identified. Human blood was detected at site A21, only 33 m from the nearest inhabited household.

**Table 1. Number of triatomines collected per landscape site and year, infection with *T. cruzi* or *T. rangeli*, and associated landscape characteristics, Cruzeiro do Sul, Acre state, Brazil.**

| Site[a] | Year | Triatomines collected | *T. cruzi*-positive | *T. rangeli*-positive | Forest cover (3 km², %) | Distance to household (m) |
|---|---|---|---|---|---|---|
| A1 | 2022 | 0 | 0 | 0 | 52.23 | 44.16 |
| A3 | 2022 | 0 | 0 | 0 | 31.60 | 326.05 |
| A4 | 2022 | 10 | 2 | 1 | 64.73 | 482.39 |
| A5 | 2022 | 1 | 0 | 0 | 10.54 | 262.70 |
| A6 | 2022 | 0 | 0 | 0 | 45.78 | 73.97 |
| A7 | 2022 | 5 | 5 | 0 | 4.18 | 98.85 |
| A10 | 2022 | 0 | 0 | 0 | 29.35 | 115.73 |
| A11 | 2022 | 0 | 0 | 0 | 60.32 | 219.62 |
| A14 | 2022 | 0 | 0 | 0 | 65.41 | 66.33 |
| A15 | 2022 | 0 | 0 | 0 | 25.41 | 85.77 |
| A16 | 2022 | 0 | 0 | 0 | 48.98 | 30.83 |
| A17 | 2022 | 0 | 0 | 0 | 54.55 | 152.09 |
| A18 | 2022 | 0 | 0 | 0 | 45.18 | 251.16 |
| A19 | 2022 | 1 | 1 | 0 | 13.62 | 177.88 |
| A20 | 2022 | 6 | 0 | 0 | 18.51 | 251.57 |
| A1 | 2024 | 0 | 0 | 0 | 47.13 | 227.59 |
| A3 | 2024 | 1 | 0 | 0 | 35.55 | 113.75 |
| A4 | 2024 | 1 | 0 | 0 | 26.99 | 228.64 |
| A5 | 2024 | 4 | 1 | 0 | 12.22 | 205.74 |
| A6 | 2024 | 2 | 0 | 0 | 41.22 | 245.68 |
| A7 | 2024 | 10 | 3 | 0 | 6.08 | 261.77 |
| A14 | 2024 | 3 | 0 | 3 | 64.12 | 210.02 |
| A20 | 2024 | 9 | 1 | 4 | 19.21 | 335.25 |
| A21 | 2024 | 2 | 0 | 0 | 58.32 | 33.01 |
| A22 | 2024 | 0 | 0 | 0 | 26.46 | 252.17 |

[a]Landscapes 02, 08, 09, 12, and 13 contained no suitable palms in 2022. In 2024, we repeated collections at sites where suitable palms were available and included landscapes A21 and A22.

To contextualize variation in triatomine occurrence and infection, we examined fine-scale landscape structure and palm-to-household distances for a subset of representative sites (Fig 3). Fig 3A illustrates a relatively preserved landscape (A14, ~65% forest cover within 3 km$^2$) where triatomines infected exclusively with *T. rangeli* were detected (Table 1). Fig 3B shows a moderately deforested landscape (site A20, ~20% forest cover) in which triatomines infected with both parasites were found, with *T. rangeli* detected more frequently than *T. cruzi*. In contrast, in the most deforested landscapes illustrated in Fig 3C and Fig 3D (sites A5 and A7, ~10% forest cover), only *T. cruzi*-infected triatomines were detected.

Species-specific models revealed contrasting responses to landscape structure (Table 2). For *T. cruzi*, forest cover within the 3 km$^2$ landscape unit had a significant negative effect on infection probability (posterior mean = −1.27; 95% credible interval: −2.52 to −0.03), indicating higher infection risk in more deforested landscapes. In addition, the interaction between forest cover and palm-household distance was positive and strongly supported (posterior mean = 1.02; 95% credible interval: 0.31 to 1.79), demonstrating that the effect of distance depended on the surrounding forest context. In

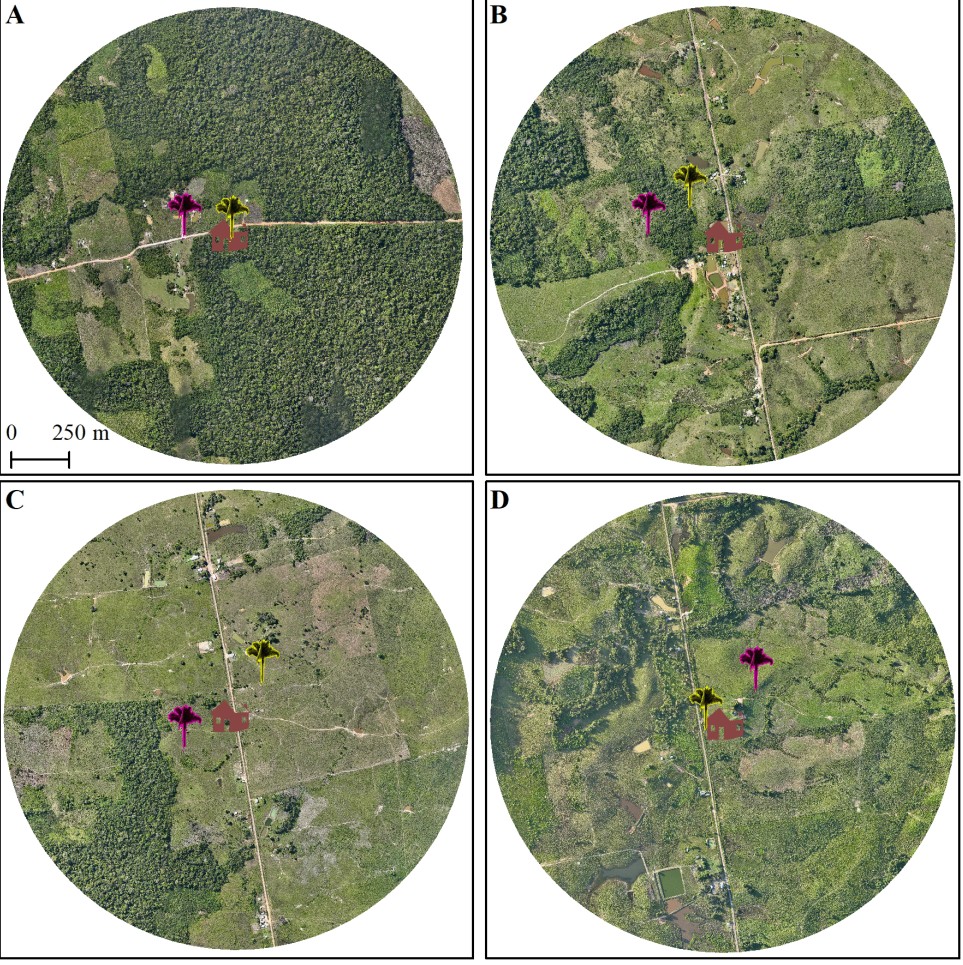

**Fig 3. Fine-scale landscape structure and palm-household distances at representative sampling sites in Cruzeiro do Sul, Acre, Brazil.** Panels show four sites across a gradient of forest cover within 3 km$^2$ landscape units: **(A)** A14 (~65% forest cover), where only *T. rangeli*-infected triatomines were detected; **(B)** A20 (~20% forest cover), with both *T. rangeli* and *T. cruzi* infections; and **(C–D)** A5 and A7 (~10% forest cover), where only *T. cruzi*–infected triatomines were detected. Yellow palm icons indicate palm trees sampled in 2022, and pink icons indicate those sampled in 2024; selected households are shown in brown. Maps were produced by the authors in ArcMap 10.8.2 using orthomosaics generated from original drone imagery (DJI Mavic 3) collected during fieldwork and processed by the authors.

 

**Table 2. Effects of landscape structure on triatomine infection probability.**

| Covariate[a] | Posterior mean | 95% credible interval | Bayesian *p*-values |
|---|---|---|---|
| *T. cruzi* | | | |
| Distance to household | −0.45 | −1.46, 0.55 | 0.19 |
| Forest cover (3 km²) | −1.27 | −2.52, −0.03 | **0.028** |
| Distance x Forest cover | 1.02 | 0.31, 1.79 | **0.0026** |
| *T. rangeli* | | | |
| Distance to household | 0.92 | −0.49, 2.40 | 0.10 |
| Forest cover (3 km²) | 0.80 | −0.95, 2.44 | 0.17 |
| Distance x Forest cover | −0.43 | −1.69, 0.96 | 0.25 |

[a]Infection was modeled as the proportion of infected triatomines per site-year, with forest cover (% within 3 km²) and palm-household distance as covariates; site was included as a random intercept.

contrast, for *T. rangeli*, none of the evaluated covariates exhibited supported effects, as all credible intervals overlapped zero.

Model-based predictions indicate that *T. cruzi* infection probability increases markedly under conditions of low forest cover, with the strongest effects observed at shorter palm-household distances (Fig 4).

## Discussion

In this study, we show that landscape structure modulates infection patterns of palm-dwelling triatomines in the south-western Amazon, with contrasting responses between *T. cruzi* and *T. rangeli* infection. By explicitly integrating fine-scale palm-household distances with landscape-level forest cover, our analysis provides new evidence that spatial context shapes parasite-specific transmission dynamics within sylvatic palm systems. Consistent with our initial hypothesis, *T. cruzi* infection was more frequent in deforested landscapes and in palms located closer to human dwellings. In contrast, *T. rangeli* showed no clear association with forest cover or palm-household distance, suggesting greater ecological stability

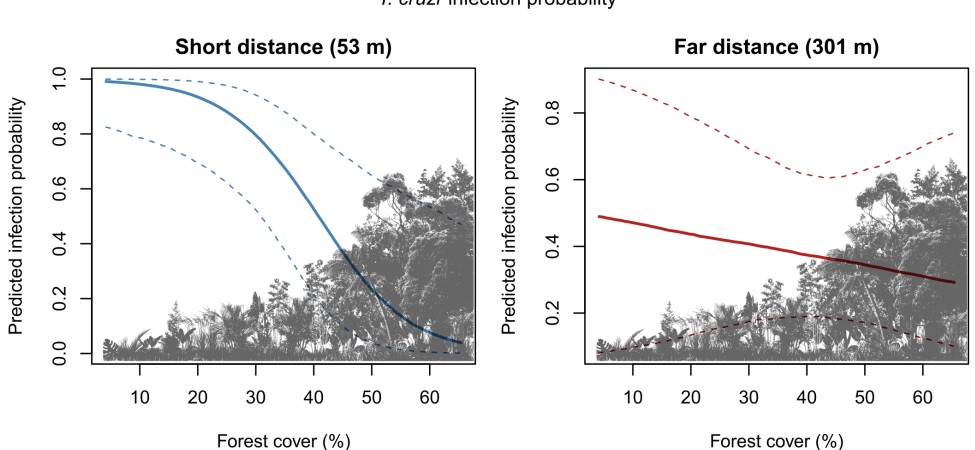

**Fig 4. Predicted probability of *T. cruzi* infection in *Rhodnius* across a gradient of forest cover (%) at short and long palm-household distances.** Short and far distances correspond to the 10th and 90th percentiles of palm-household distances (i.e., 53 and 301 meters, respectively). Solid lines represent posterior medians and dashed lines indicate 95% credible intervals.

across landscape gradients. Together, these findings indicate that deforestation and human proximity selectively favor *T. cruzi* transmission within sylvatic palm-based triatomine systems.

The negative association between forest cover and *T. cruzi* infection probability supports the idea that habitat degradation reshapes host-vector-parasite interactions in Amazonian landscapes [23]. Forest loss is known to reduce vertebrate host diversity while favoring generalist and synanthropic mammals, including marsupials, which are key reservoirs of *T. cruzi* [49]. Consistent with this pattern, marsupial blood meals were frequently detected at sites where *T. cruzi* infection occurred in our study, reinforcing their role in sustaining transmission within disturbed palm habitats. Previous studies in the Amazon have also reported higher triatomine abundance in deforested landscapes than in continuous forest [50]. These findings suggest that deforestation not only alters host community composition but may also increase vector presence and infection risk, contributing to enhanced *T. cruzi* circulation in human-modified environments [38].

The significant interaction between forest cover and palm-household distance indicates that proximity to human dwellings does not operate independently of landscape context [51]. Short palm-household distances were associated with higher *T. cruzi* infection probability primarily in deforested landscapes, whereas this effect was attenuated under higher forest cover. This pattern is consistent with previous studies showing that palms located closer to houses often harbor higher numbers of triatomines per tree, particularly in disturbed settings, which may further amplify infection risk near dwellings [52,53]. Forested environments may buffer the influence of human proximity by maintaining more complex sylvatic transmission cycles, while deforested landscapes facilitate spillover processes at the sylvatic-domestic interface [54].

In contrast to *T. cruzi*, we did not detect associations between landscape metrics and *T. rangeli* infection probability. This parasite is considered non-pathogenic to humans and is commonly associated with stable sylvatic transmission cycles involving palm-dwelling triatomines and forest vertebrates [3]. Its apparent insensitivity to forest cover and palm-household distance may reflect broader host use, lower pathogenic costs to vectors, or reduced dependence on anthropized environments [4].

The detection of human blood meals, including in a fourth-instar nymph collected only 33 m from an inhabited household, underscores the permeability of sylvatic palm systems to human exposure [16]. Because nymphal stages are wingless and have limited dispersal capacity, this finding strongly suggests local host-vector contact rather than long-distance movement, indicating that feeding on humans can occur within or immediately adjacent to dwellings [55,56]. The concurrent detection of *T. cruzi* DTU TcI, the lineage most commonly associated with sylvatic transmission cycles and human infections in the Amazon, further highlights the epidemiological relevance of these encounters [57–59]. TcI is frequently implicated in outbreaks of orally transmitted ACD and is characterized by broad host plasticity, enabling its persistence across gradients of forest disturbance and host community change [60,61].

Although palm-associated triatomines are traditionally regarded as strictly sylvatic, our results add to growing evidence that deforestation increases trophic connectivity between forest habitats and human-modified environments [62,63]. In this context, palms retained near dwellings in deforested settings may function as ecological stepping stones, bringing sylvatic vectors, wildlife hosts, and humans into closer and more frequent contact [38]. The presence of TcI-infected triatomines feeding on humans suggests that anthropized landscapes may facilitate the spillover of enzootic transmission cycles into peridomestic contexts without the need for stable household colonization [64–66]. This process is particularly relevant in Amazonian settings, where palms are commonly preserved for cultural, economic, or subsistence purposes and often occur near houses and food preparation areas, creating repeated opportunities for vector contact, contamination of food substrates, and oral transmission [8,29,67–69]. Rather than advocating broad wildlife removal, these findings highlight the potential value of targeted environmental management, such as the spatial planning, selective management, or substitution of palm trees near households, as complementary strategies to reduce human exposure at the sylvatic-human interface.

Although our findings reveal clear landscape-mediated patterns in triatomine infection, sample sizes were constrained by the low density and patchy distribution of palm-associated triatomines, limiting inference to infection probability. In

addition, sites with higher forest cover may include secondary or regenerating forests; variation in forest age and tree species composition, not captured in our analysis, could influence palm availability, host assemblages, and *T. cruzi* transmission risk. Future studies integrating longitudinal sampling, tree species composition matrices, and host community data will be essential to clarify the mechanisms linking deforestation and parasite transmission in Amazonian palm systems.

## Conclusions

Landscape structure strongly influences parasite-specific infection patterns in palm-dwelling triatomines in the southwestern Amazon. *T. cruzi* infection was more frequent in deforested landscapes and further modulated by proximity to human dwellings. The detection of *T. cruzi* DTU TcI and human blood meals highlights the permeability of sylvatic palm systems and the epidemiological relevance of landscape disturbance. These findings indicate that deforestation reshapes host-vector-parasite interactions, increasing spillover risk at the sylvatic-human interface independently of documented triatomine domiciliation.

## Supporting information

**S1 Fig. *Rhodnius* triatomine specimen collected in the field.**
(JPEG)

**S1 Table. Molecular detection of *T. cruzi* and *T. rangeli* and identification of blood meal sources in *Rhodnius* spp. collected from palm trees across landscape sites in Cruzeiro do Sul, Acre state, Brazil, in 2022 and 2024.**
(DOCX)

**S1 Video. Field collection of palm-dwelling triatomines from palm bracts.**
(MP4)

## Acknowledgments

We extend our gratitude to the staff of the Municipal Health Secretariat of Cruzeiro do Sul and the Acre State Health Services, with special acknowledgment to Antônio de Oliveira, community health agent, and Helio Cameli and Leonisio Mendonça, administrative officers, for their invaluable support and enduring collaboration in this research. Lastly, we sincerely thank the rural villagers for their continued participation and contributions to this study.

## Author contributions

**Conceptualization:** Gabriel Z. Laporta, Paula R. Prist, Denis Valle.

**Data curation:** Gabriel Z. Laporta, Fernanda P. Madeira, Marcia A. Sperança.

**Formal analysis:** Gabriel Z. Laporta, Roberto C. Ilacqua, Denis Valle.

**Funding acquisition:** Gabriel Z. Laporta, Melissa S. Nolan, Andreia F. Brilhante, Marcia A. Sperança.

**Investigation:** Leandro J. Ramos, Carla M. Santana, Wandercleyson U. Abreu, Roberto C. Ilacqua, Andreia F. Brilhante, Fredy Galvis-Ovallos, Marcia A. Sperança.

**Methodology:** Leandro J. Ramos, Carla M. Santana, Wandercleyson U. Abreu, Roberto C. Ilacqua, Andreia F. Brilhante, Fernanda P. Madeira, Fredy Galvis-Ovallos, Marcia A. Sperança.

**Project administration:** Fernanda P. Madeira, Marcia A. Sperança.

**Resources:** Roberto C. Ilacqua, Melissa S. Nolan, Dionatas U. O. Meneguetti, Marcia A. Sperança.

**Supervision:** Melissa S. Nolan, Paula R. Prist, Denis Valle.

**Validation:** Gabriel Z. Laporta, Melissa S. Nolan, Dionatas U. O. Meneguetti, Paula R. Prist, Denis Valle, Marcia A. Sperança.

**Visualization:** Gabriel Z. Laporta, Melissa S. Nolan, Andreia F. Brilhante, Fernanda P. Madeira, Dionatas U. O. Meneguetti, Fredy Galvis-Ovallos, Paula R. Prist, Denis Valle, Marcia A. Sperança.

**Writing – original draft:** Gabriel Z. Laporta.

**Writing – review & editing:** Gabriel Z. Laporta, Leandro J. Ramos, Carla M. Santana, Wandercleyson U. Abreu, Roberto C. Ilacqua, Melissa S. Nolan, Andreia F. Brilhante, Fernanda P. Madeira, Dionatas U. O. Meneguetti, Fredy Galvis-Ovallos, Paula R. Prist, Denis Valle, Marcia A. Sperança.

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
