## [Decision Letter · Decision Letter 0]

9 Apr 2026

PONE-D-26-09584Deforestation and human proximity influence Trypanosoma cruzi infection in palm-dwelling triatominesPLOS One

Dear Dr. Laporta,

Thank you for submitting your manuscript to PLOS ONE. After careful consideration, we feel that it has merit but does not fully meet PLOS ONE’s publication criteria as it currently stands. Therefore, we invite you to submit a revised version of the manuscript that addresses the points raised during the review process.

We look forward to receiving your revised manuscript.

Kind regards,

Roberto Magalhães Saraiva, MD, PhD

Academic Editor

PLOS One

Journal Requirements:

4. Please note that funding information should not appear in any section or other areas of your manuscript. We will only publish funding information present in the Funding Statement section of the online submission form. Please remove any funding-related text from the manuscript.

6. We note that Figure 2 and 3 in your submission contain map images which may be copyrighted. All PLOS content is published under the Creative Commons Attribution License (CC BY 4.0), which means that the manuscript, images, and Supporting Information files will be freely available online, and any third party is permitted to access, download, copy, distribute, and use these materials in any way, even commercially, with proper attribution. For these reasons, we cannot publish previously copyrighted maps or satellite images created using proprietary data, such as Google software (Google Maps, Street View, and Earth). For more information, see our copyright guidelines: http://journals.plos.org/plosone/s/licenses-and-copyright.

1. You may seek permission from the original copyright holder of Figure 2 and 3 to publish the content specifically under the CC BY 4.0 license.

Reviewer's Responses to Questions

**Comments to the Author**

1. Is the manuscript technically sound, and do the data support the conclusions?

Reviewer #1: Yes

2. Has the statistical analysis been performed appropriately and rigorously? 

Reviewer #1: N/A

3. Have the authors made all data underlying the findings in their manuscript fully available?

Reviewer #1: Yes

4. Is the manuscript presented in an intelligible fashion and written in standard English?

Reviewer #1: Yes

5. Review Comments to the Author

Reviewer #1: The manuscript is clearly written and well organized. The objectives, methodology, and results are presented in a coherent and understandable way. The conclusions are supported by the data presented. I do not have major comments that would require substantial revision.

Additional comments:

INTRODUCTION: Maybe add some more information regarding T. rangeli, explaining it is a species that only infects and affects Rhodnius triatomines (as you did in the discussion), and perhaps add the implications that T. rangeli has for diagnosis when it is present along with T. cruzi in samples, causing cross-reactivity and misdiagnosis, or that it can be implicated in false-positive results according to the diagnostic technique used. The above could help provide more support for the decision to also include T. rangeli in the study.

FIGURE 2: In the top left map, consider adding a small inset map showing the location of Brazil within the Americas.

References appear to be in a different font than the rest of the manuscript.

6. PLOS authors have the option to publish the peer review history of their article (what does this mean?). If published, this will include your full peer review and any attached files.

Reviewer #1: No

---

## [Author Response · Author response to Decision Letter 1]

14 Apr 2026

Response to the Editor and Reviewer

Manuscript ID: PONE-D-26-09584

Title: Deforestation and human proximity influence Trypanosoma cruzi infection in palm-dwelling triatomines

Dear Dr. Saraiva and Reviewers,

We would like to sincerely thank the Academic Editor and the Reviewer for their careful evaluation of our manuscript and for the constructive and valuable comments. We are pleased that the reviewer found the study clearly written, methodologically sound, and scientifically relevant. We have carefully addressed all points raised and revised the manuscript accordingly.

Below we provide a detailed, point-by-point response to all comments.

Response to the Academic Editor’s and Journal Requirements

1. Figures 2 and 3 – Copyright and licensing concerns

Comment: Figures may contain copyrighted map or satellite data and may not comply with CC BY 4.0 requirements.

Response:

We thank the editor for raising this important issue. We would like to clarify that Figures 2 and 3 were entirely produced by the authors using non-proprietary and/or original data sources.

Figure 2 was generated using publicly available datasets, including:

Administrative boundaries from IBGE

Land use/land cover data from MapBiomas

Field-collected GPS coordinates

Figure 3 was produced using orthomosaic imagery derived from drone surveys (DJI Mavic 3) conducted by the authors during fieldwork. These images were processed and analyzed entirely by the authors.

Importantly, no proprietary mapping platforms (e.g., Google Maps, Google Earth, or similar services) were used. Therefore, the figures are fully compatible with the CC BY 4.0 license.

In addition, we have revised the figure captions to explicitly state data sources and confirm authorship and originality.

2. ORCID requirement

Response:

We confirm that the ORCID iD of the corresponding author has been validated in the Editorial Manager system.

3. Funding statement inconsistency

Response:

We thank the editor for identifying this issue. We have carefully reviewed the funding information and corrected inconsistencies between the Funding Information section and the Financial Disclosure statement.

4. Funding information in manuscript text

Response:

We confirm that all funding-related information has been removed from the manuscript body and is now included only in the designated Funding Statement section.

5. Ethics statement placement

Response:

We have revised the manuscript to ensure that the ethics statement appears exclusively in the Methods section, as requested. Any duplicate mentions elsewhere in the manuscript have been removed.

6. References formatting

Response:

We have carefully reviewed the reference list and corrected formatting inconsistencies, including font differences and style alignment with PLOS ONE requirements. The revised manuscript now follows the journal’s formatting guidelines consistently.

Response to Reviewer #1

We thank Reviewer #1 for the positive evaluation and constructive suggestions.

Comment 1: Inclusion of additional information on Trypanosoma rangeli in the Introduction

Response:

We appreciate this insightful suggestion. We have expanded the Introduction to include additional background on Trypanosoma rangeli, highlighting:

Its association with Rhodnius spp. triatomines

Its non-pathogenic nature to humans

Its epidemiological relevance due to co-circulation with T. cruzi

Its potential to cause cross-reactivity in serological and molecular diagnostic assays, which may lead to false-positive results depending on the diagnostic method used

This addition strengthens the rationale for including T. rangeli in our study design.

Comment 2: Figure 2 – add inset map showing Brazil within the Americas

Response:

We thank the reviewer for this helpful suggestion. The inset map indicating the location of Brazil within the Americas was already included in the original Figure 2. However, to improve clarity and visibility, we have revised the figure by enhancing the inset map resolution and improving its visual prominence, as well as better delineating the study area within Brazil.

Comment 3: Reference formatting inconsistency

Response:

We thank the reviewer for identifying this issue. We have corrected the reference formatting to ensure consistency throughout the manuscript according to PLOS ONE style requirements.

We sincerely appreciate the time and effort of the Editor and Reviewer in evaluating our manuscript. We believe that the revisions made have significantly improved the clarity, rigor, and compliance of the manuscript with PLOS ONE requirements.

We hope that the revised version is now suitable for publication in PLOS ONE.

Sincerely,

Gabriel Laporta and co-authors

---

## [Decision Letter · Decision Letter 1]

29 Apr 2026

Deforestation and human proximity influence Trypanosoma cruzi infection in palm-dwelling triatomines

PONE-D-26-09584R1

Dear Dr. Laporta,

We’re pleased to inform you that your manuscript has been judged scientifically suitable for publication and will be formally accepted for publication once it meets all outstanding technical requirements.

Kind regards,

Roberto Magalhães Saraiva, MD, PhD

Academic Editor

PLOS One

Additional Editor Comments (optional):

Reviewers' comments:

Reviewer's Responses to Questions

**Comments to the Author**

1. If the authors have adequately addressed your comments raised in a previous round of review and you feel that this manuscript is now acceptable for publication, you may indicate that here to bypass the “Comments to the Author” section, enter your conflict of interest statement in the “Confidential to Editor” section, and submit your "Accept" recommendation.

Reviewer #1: All comments have been addressed

2. Is the manuscript technically sound, and do the data support the conclusions?

Reviewer #1: Yes

3. Has the statistical analysis been performed appropriately and rigorously? 

Reviewer #1: Yes

4. Have the authors made all data underlying the findings in their manuscript fully available?

Reviewer #1: Yes

5. Is the manuscript presented in an intelligible fashion and written in standard English?

Reviewer #1: Yes

6. Review Comments to the Author

Reviewer #1: (No Response)

7. PLOS authors have the option to publish the peer review history of their article (what does this mean?). If published, this will include your full peer review and any attached files.

Reviewer #1: No

---

## [Editor Report · Acceptance letter]

PONE-D-26-09584R1

PLOS One

Dear Dr. Laporta,

I'm pleased to inform you that your manuscript has been deemed suitable for publication in PLOS One. Congratulations! Your manuscript is now being handed over to our production team.

Kind regards,

on behalf of

Dr. Roberto Magalhães Saraiva

Academic Editor

PLOS One